# Application and Monitoring of Oxidative Alginate–Biocide Hydrogels for Two Case Studies in "The Sassi and the Park of the Rupestrian Churches of Matera"

Francesco Gabriele [1], Laura Bruno [2], Cinzia Casieri [1], Roberta Ranaldi [2], Lorenza Rugnini [2] and Nicoletta Spreti [1,*]

1 Department of Physical and Chemical Sciences, University of L'Aquila, Via Vetoio—Coppito, I-67100 L'Aquila, Italy; francesco.gabriele@graduate.univaq.it (F.G.); cinzia.casieri@aquila.infn.it (C.C.)
2 LBA—Laboratory of Biology of Algae, Department of Biology, University of Rome "Tor Vergata", Via Cracovia 1, I-00133 Rome, Italy; laura.bruno@uniroma2.it (L.B.); ranaldi.roberta@gmail.com (R.R.); lorenza.rugnini@uniroma2.it (L.R.)
* Correspondence: nicoletta.spreti@univaq.it

**Abstract:** The removal of biological colonization on building materials of cultural heritage is a difficult challenge, as the treatment must completely eliminate the biological patina without altering the treated substrate and possibly delaying new colonization. With the aim of searching for systems to minimize the biocide impact on the substrate, the environment and the operators, different alginate–oxidizing biocide hydrogels were previously tested and optimized in the laboratory and here selected for application in situ. The churches "San Pietro Barisano" and "Madonna dei Derelitti", located in the Sassi of Matera (UNESCO World Heritage Site in Basilicata region, Italy), were chosen as case studies. They differ in terms of both the environmental conditions and the microorganisms responsible for colonization. Colorimetric measurements and microscopic investigation proved the efficacy of biocide hydrogels in removing biopatinas and in restoring the original chromaticity of the selected treated surfaces of both sites. After the biocidal treatments, new protective acrylic coatings were applied to prevent recolonization and minimize the loss of material grains. Samples collected, immediately after and two years later, established the absence of biological colonization, demonstrating the long-term efficacy of the proposed restoration protocol.

**Keywords:** calcareous stone; biodeterioration; phototrophic biofilms; biocide; hydrogel; cleaning; protective coating; rupestrian church; Sassi of Matera; long-term efficacy

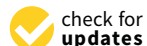



## 1. Introduction

In recent years, the biodeterioration of stone artworks has received notable attention; this phenomenon, currently ascribed to both natural and anthropogenic causes along with the bioreceptivity of the substratum [1–3] and the type of colonizing microorganisms [4–7], greatly influences their decay. In particular, the chemical composition, roughness, capillarity and superficial porosity of the constitutive material can affect the colonization of different microorganisms, such as green algae, diatoms, cyanobacteria, bacteria, lichens, fungi and mosses.

Microbial colonization of stone works of art has long been considered only as an aesthetic form of deterioration due to the classic chromatic variations associated with the microorganisms [7,8]. Unfortunately, biological colonization not only affects the appearance of cultural heritages but can also induce several damages to the material surface and in the inner porous structure [9,10]. Photoautotrophic microorganisms (green microalgae and cyanobacteria), which do not require organic material for growth, are often the first colonizers of stone surfaces. Through their metabolic activity, the stone materials are enriched with nutrients that can lead to the growth of heterotrophic microorganisms (bacteria and

fungi). These communities of microorganisms, embedded in a polymeric matrix [11,12], constitute a biofilm, responsible for the biodeterioration of stone cultural heritage.

The cleaning process of works of art is always a difficult challenge, as the treatment must completely remove the biological patina without altering the treated substrate and possibly delaying new colonization. For this reason, after cleaning, consolidation and protection treatments should be performed.

The most common methods employed for the removal of biofilms from stone heritages can be divided into mechanical (abrasion, scraping and scrubbing), physical (microwave, laser and ultraviolet radiation), chemical (natural and synthetic biocides) and biological (bacteria and enzymes) techniques [9,10,13–15]. Stone surfaces have traditionally been treated with chemical biocides, although most of them are ineffective in removing microorganisms; moreover, they can cause degradative or corrosive effects on the substrate surface and are often harmful to human health and the environment.

The encapsulation of biocides in inert matrices has proved to be a very useful means to minimize the drawbacks ascribed to chemical treatments, since both the amount of the biocide and the penetration of by-products can be reduced. In addition, more efficient removal of microorganisms can be achieved by a soft mechanical action [16–18].

Recently, in this contest, we have reported part of the multidisciplinary results of a Smart Cities and Communities and Social Innovation project (SCN_00520), supported by MIUR and titled "Product and process innovation for the maintenance, conservation and sustainable restoration of cultural heritage". Two polysaccharide–biocide hydrogels, based on ionically crosslinked alginate and containing two oxidizing agents, were tested in the laboratory for cleaning limestone subjected to biological attack. Alginate hydrogels containing calcium hypochlorite, as both biocide and crosslinker (BH_1) [19], and sodium dichloroisocyanurate (NaDCC), as a reservoir of hypochlorous acid (BH_2) [20], proved effective in removing biofouling on standardized Lecce stone specimens, chosen as a material widely used in historic Baroque buildings in southern Italy. Optical microscopy, colorimetry and NMR relaxometry revealed the efficacy of both hydrogels without altering the properties of the treated specimens.

Specifically, the here-presented data, belonging to the same Smart City project, concern the in situ application and monitoring of the newly developed biocide hydrogels and protective–consolidant products on stone buildings belonging to the "The Sassi and the Park of the Rupestrian Churches of Matera", located in the southern Italian region of Basilicata, included by UNESCO among the World Heritage Sites in 1993. Two different sites were selected: the church "San Pietro Barisano" (Figure 1a), one of the largest and most visited rupestrian churches in the city of Matera, and the church "Madonna dei Derelitti" (Figure 1b), located on the side of the Gravina opposite the one on which the Sassi districts stand and currently closed to visitors.

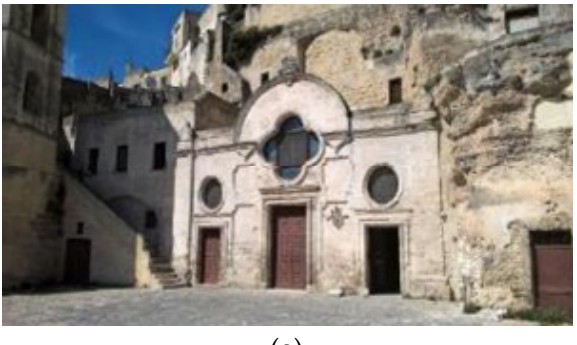
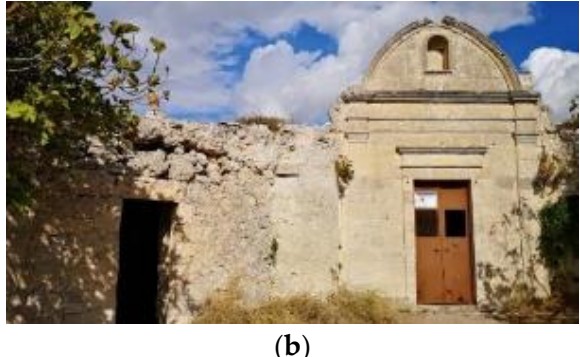

(**a**)          (**b**)

**Figure 1.** Monumental buildings belonging to the "The Sassi and the Park of the Rupestrian Churches of Matera", located in the southern Italian region of Basilicata. (**a**) "San Pietro Barisano" church, a typical example of the rock architecture of the Sass; (**b**) "Madonna dei Derelitti" church, located on the Gravina side.

The optimized BH_1 was applied on two different areas located in the hypogean level of the church San Pietro Barisano, once dedicated to the draining of corpses, called "*Putridarium*", while three different areas of intervention, identified within the church Madonna dei Derelitti according to the type of biodeterioration, were treated with BH_2. After the cleaning operations, new protective–consolidant products, developed by Icap Leather Chem SpA (Milan, Italy), were selected through a comparison study with others of different brands, already on the market [21].

Some samples were collected from the surfaces two years after the treatments to monitor their effectiveness over time.

## 2. Materials and Methods

### 2.1. Areas of Intervention

As part of the aforementioned Smart City project, thanks to the adhesion of the competent authorities (municipality, superintendency and diocese of Matera), we had the opportunity to test our hydrogel formulations in situ in the two selected sites.

#### 2.1.1. Site 1: San Pietro Barisano Church

Two areas inside the *Putridarium* of San Pietro Barisano (Italy), both illuminated with an artificial light source and intensively covered by microbial biofilms, were selected: SPB-1, a section of a column of $0.4 \times 1.0$ (l × h) m², located in the area where there are masonry seats on which the bodies of decomposing monks were temporarily placed, and SPB-2, the right wall of $1.0 \times 1.0$ (l × h) m² of a small niche of dimensions $1.5 \times 0.9 \times 2.0$ (l × w × h) m³ (Figure 2a).

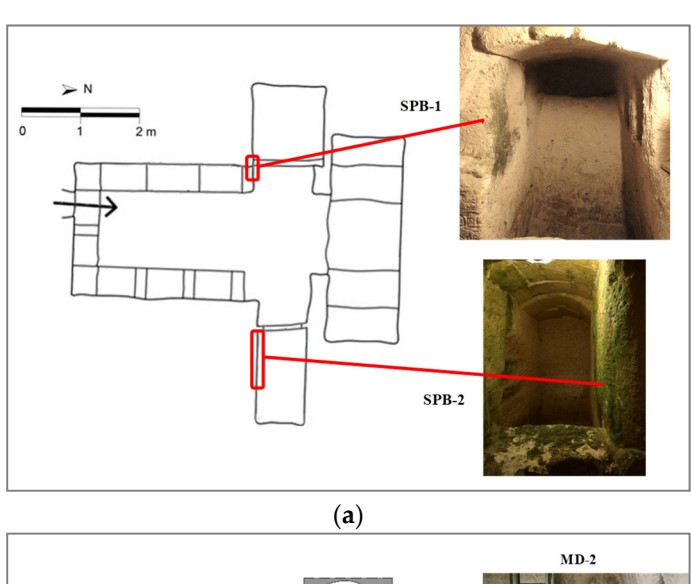

(a)

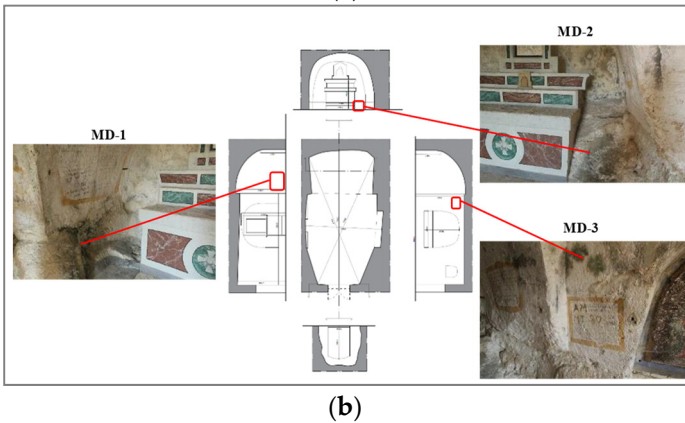

(b)

**Figure 2.** Map of the interior of the churches San Pietro Barisano (**a**) and Madonna dei Derelitti (**b**) and photos of the areas subjected to the cleaning procedure.

### 2.1.2. Site 2: Madonna dei Derelitti Church

Three different areas inside the church were selected: two areas to the left (MD-1) and right (MD-2) of the altar, entirely excavated in the limestone, and a small, colonized area (MD-3), located in the right wall, probably subjected to whitewashing, as in the lower part there is a fresco (Figure 2b).

### 2.2. Environmental Parameters

During the sampling, the environmental parameters were recorded at each sampling site. The humidity and temperature of the wall surface were measured respectively by means of a contact hygrometer, model Hydromette BL UNI 11 (GANN, Gerlingen, Germany), equipped with an active electrode B 55 BL, and a thermographic camera, model C2 (FLIR, Wilsonville, OR, USA).

Ambient temperature (°C) and relative humidity (RH%) were measured using a thermo-hygrometer model HI 18564 (Hanna Instruments, Woonsocket, RI, USA). Irradiance was measured using an LI185 portable radiometer (LI-COR, Lincoln, NE, USA) equipped with a quantum sensor (LI-190SB) for the determination of the photosynthetic photon flux density (PPFD) in μmol photons $m^{-2} \cdot s^{-1}$.

### 2.3. Sampling of Biofilms and Their Characterization

Samples of biofilms were collected from the walls by means of scraping with sterile scalpels or with the nondestructive sampling method of the adhesive tape strips, a consolidated method to collect samples in situ [22]. To characterize the phototrophs in the biofilms, samples were observed with a Zeiss AxioScope light microscope with a 40× objective equipped with a digital camera (Canon EOS 600D—Canon S.P.A., Milan, Italy) for image acquisition. The adhesive tape strips were also observed with an FV1000 confocal laser scanning microscope (CLSM) (Olympus Corp., Tokyo, Japan) with a 60× objective using the autofluorescence channels for chlorophyll *a* and phycobiliproteins (excitation 488, 543 and 635 nm; emission 520, 572 and 688 nm). Three-dimensional images were constructed from a series of 2D cross-sectional images (x–y plane) that were captured at 0.5 μm intervals along the z-axis using IMARIS 6.2.0 software (Bitplane AG Zurich, Zürich, Switzerland). Scanning Electron Microscopy (SEM)/Energy Dispersive Spectroscopy (EDS) analyses were performed using Zeiss GeminiSEM 500 (Jena, Germany) equipped with EDS Oxford Aztec Energy with INCA X-ACT detector (8.5 mm working distance and 15 KeV accelerating voltage, Oxford Instruments, Abingdon-on-Thames, UK).

To further evaluate the possible presence in situ of few cells of colonizing phototrophs, still alive after the treatment and not detectable by microscope observation, small pieces of the adhesive tape strips from the samples collected in both churches were cut and then put on the surface of agarized BG11 medium.

### 2.4. Macroscopic and Microscopic Observations

Macroscopic observations were performed by visual inspection and photographic recording using a Canon EOS 1300D digital camera (Canon S.P.A., Milan, Italy) before the cleaning treatment, shortly after the cleaning treatment and 2 years after cleaning treatment. Optical images of noncolonized, colonized and cleaned surfaces were acquired in situ using a FlipView 5MP portable digital microscope (Celestron, Torrance, CA, USA).

### 2.5. Colorimetric Measurements

To evaluate the degree of biodegradation of stone materials and the effectiveness of hydrogel treatments to restore the original chromaticity, colorimetric measurements were performed before the treatment, immediately after and then after two years. The Sama Tools SA230 portable colorimeter (Viareggio, Italy) was used for colorimetric analyses; measurements were performed in SCE mode with an 8° standard observer, light D65 (average daylight, including the UV region, with the relative color temperature of 6504 K). The instrument was calibrated with the white reference. The colorimetric parameters in CIELAB

color space were determined according to UNI EN15886:2010 [23]. The color modification ($\Delta E^*$) was calculated using the following relation: $DE^* = (DL^{*2} + Da^{*2} + Db^{*2})^{\frac{1}{2}}$.

*2.6. Preparation and Application of the Alginate–Biocide Hydrogels*

All chemicals, alginic acid sodium salt from brown algae (low viscosity), calcium hypochlorite, sodium dichloroisocyanurate and calcium chloride, were purchased from Merck (Kenilworth, NJ, USA).

The compositions of the two alginate–biocide hydrogels were previously optimized [19,20] and both formulations were already used for the cleaning of artificially colonized Lecce stones. Given the large surfaces to be cleaned, the preparation of the biocidal hydrogels was optimized to facilitate restorers for in situ application. The procedure was developed in accordance with Pantone Restauri Srl (Rome, Italy), which collaborated in the preparation, application and removal of the hydrogels.

*Preparation of BH_1.* An aqueous solution of $Ca(ClO)_2$ (2 wt.%—500 mL) in 1.5 wt.% glacial acetic acid was slowly added to an aqueous alginate solution (10 wt.%—500 mL) under vigorous stirring until a compact and homogeneous gel was obtained.

*Preparation of BH_2.* A 500 mL aqueous solution containing NaDCC (1.6 wt.%) and $CaCl_2$ (0.6 wt.%) was slowly added to an aqueous alginate solution (10 wt.%—500 mL) under vigorous stirring until a compact and homogeneous gel was obtained.

The characterization of both biocidal hydrogels, reported in Supplementary Materials (Figure S1 and Table S1), highlighted that the addition of the oxidizing agents did not significantly change the chemical–physical properties of the hydrogel.

The biocide hydrogels were applied by brush on a cotton gauze adhering to the surfaces to be treated to accomplish a homogeneous application and an easy removal; to speed up the drying of the gel, thermo-convectors, ventilators and/or infrared lamps were used. In one of the first tentative cleaning tests, the hydrogel was applied with a polystyrene net that stretched and did not adhere to the surface; consequently, the hydrogel was mostly adherent to the stone material and was difficult to remove. In this case, a second treatment had to be carried out to remove hydrogel residues: washing the surface with a saturated solution of bicarbonate and demineralized water. The washing cycles included one spray wash with the saline solution and three rinses with demineralized water.

*2.7. Protective Coatings*

Among the coatings developed by Icap Leather Chem (Milan, Italy) and previously monitored [21], the most effective ones were Polyrest AC012, Polyrest P1 and Polyrest P2, acrylic copolymers in aqueous emulsion, which slow down hydration kinetics without modifying the color of the limestone samples. After the cleaning treatment on both Site 1 and Site 2, different areas were treated with the Icap Leather Chem SpA products Polyrest AC012 and Polyrest P2. Aqueous emulsions (10%) of the protective products were laid twice on selected areas by spraying until surface saturation.

*2.8. Peeling Test*

The test was based on the use of an adhesive tape of a known area, which by mechanical action removes stone material according to the chalking of the material; by weighing, a relative chalking value is attributed [24].

**3. Results and Discussion**

The experimental phases of the application of the biocidal hydrogels were performed in the *Putridarium* of the church San Pietro Barisano in June 2019 and in the church Madonna dei Derelitti in October 2019. Evident biodeterioration processes, due to the presence of green patinas that covered large areas of the underlying walls, heavily affected both sites.

Before the cleaning treatments, surface and environmental temperature and humidity were monitored; furthermore, several biofilm samples were collected using the method of adhesive tape strips as a nondestructive sampling method [25]. Then, the selected

areas were cleaned with biocidal hydrogels and treated with protective coatings. Each phase of our intervention was accomplished by means of photographic documentation and colorimetric analysis. Finally, two years later, in July 2021, the long-term effectiveness of both cleaning and protection treatments was evaluated.

### 3.1. Site 1: The Putridarium of San Pietro Barisano Church

High relative humidity, low temperature and artificial lighting, even if low, but enough for tourist visits, characterize the peculiar environment of this hypogean site and these factors can favor the growth of photosynthetic biofilms, especially on the illuminated stone surfaces.

The environmental conditions of this site were measured: RH ranged from 75% to 86% with a temperature of 18 °C and irradiance from 1.2 to 6 µmol of photons $m^{-2} \cdot s^{-1}$. For a better characterization of the site, the relative humidity and temperature of the wall surfaces were also acquired; RH values were always above 90%, and temperatures were between 11 and 17 °C.

Before proceeding with the cleaning procedure, an apparent noncolonized area (Figure 3a) was identified within the site in order to obtain the colorimetric parameters of a reference (REF-SPB) with which to compare the results obtained on the treated areas, after each kind of intervention.

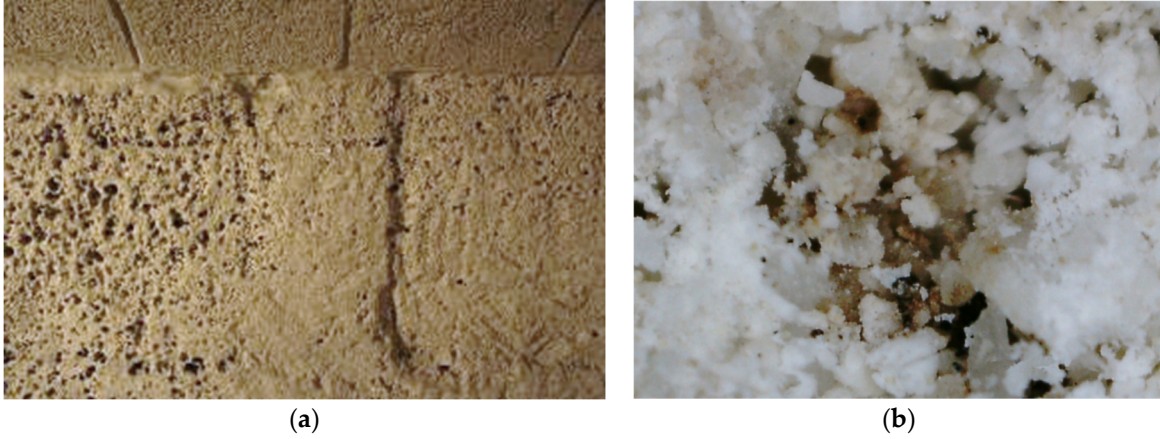

| | |
|:---:|:---:|
| (**a**) | (**b**) |

**Figure 3.** (**a**) Photo and (**b**) digital microscopy image of an uncolonized wall of the *Putridarium* in the church San Pietro Barisano selected as a reference (REF-SPB).

The selected area was then observed by digital microscopy (Figure 3b), which would seem to highlight the absence of biological patina on the surface. The biological characterization definitely confirmed the absence of microorganisms on the chosen area.

The mean color parameters of REF-SPB were L* = 72 ± 5, a* = 4.8 ± 0.9 and b* = 13 ± 2. These parameters were utilized, as described in the following, to evaluate the variation of color, ΔE*.

On the areas SPB-1 and SPB-2, colorimetric measurements were previously performed to compare the sample areas, before and after treatments, with REF-SPB.

As shown in Figure 4a, the SPB-1 area was highly colonized by deteriorating biofilms, and the stone material on which the biological patina was present showed an evident discoloration, clearly visible to the naked eye. This area was also inspected through portable microscopy, as can be seen in Figure 4d with 4× magnification.

For what concerns SPB-2, in Figure 4f, a high degree of green colonization can be observed which entirely covers the area of interest. For this area, a surface of approximately 1 m² was selected to evaluate the approximate quantity of hydrogel required for the surface unit, which was found to be ≅1 kg/m². The digital microscopy image of the biopatina on the SPB-2 area is shown in Figure 4i.

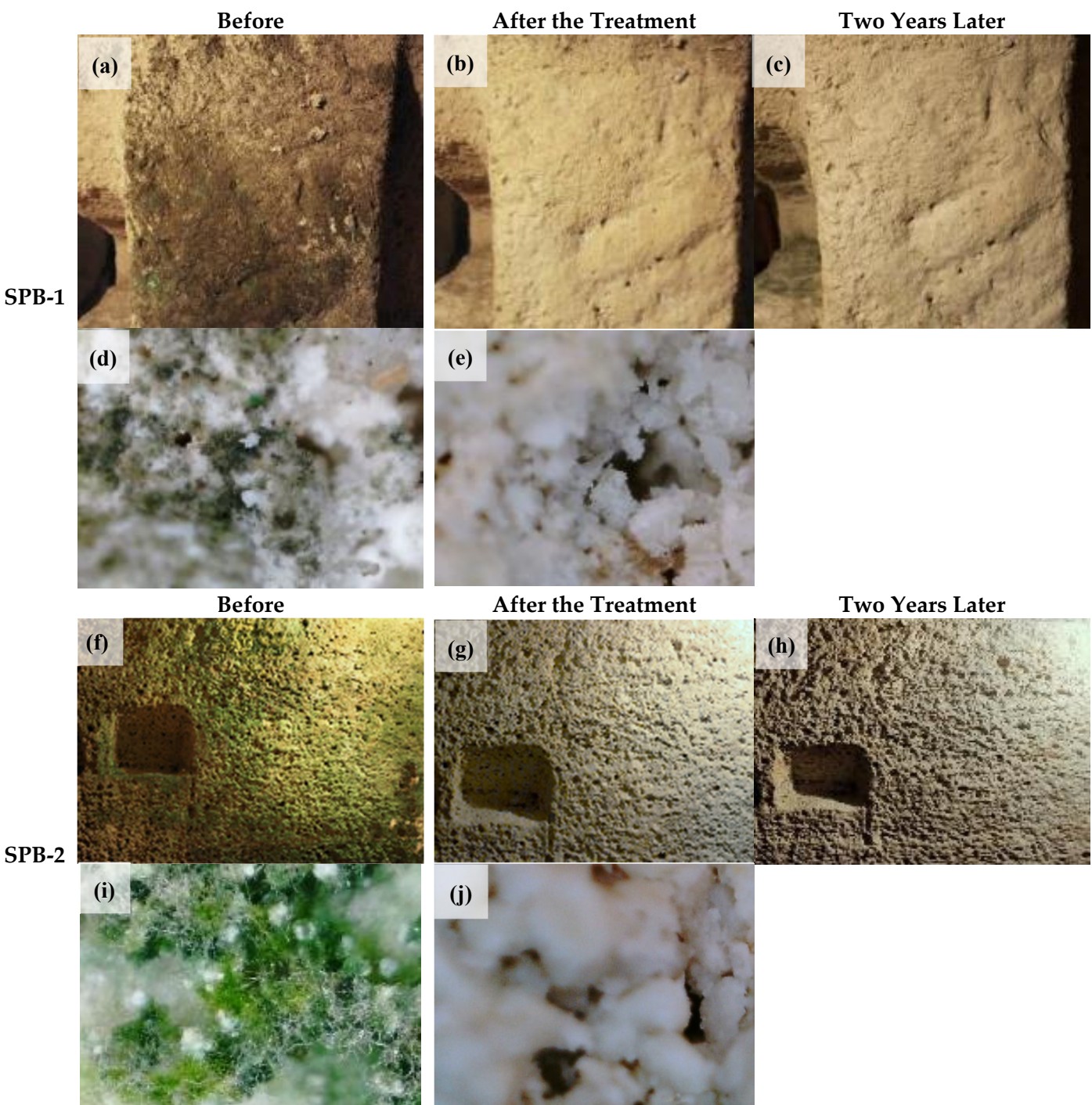

**Figure 4.** Photos of SPB-1 (**a**–**c**) and SPB-2 (**f**–**h**) and digital microscopy images with 4× magnification of SPB-1 (**d**,**e**) and SPB-2 (**i**,**j**) areas of the *Putridarium* in the church San Pietro Barisano, before the cleaning procedure (left column), shortly after (middle column) and two years after (right column) the cleaning procedure.

Samples collected in both sites before the cleaning procedure, to characterize the microorganisms composing the biopatina, were observed with a light microscope, showing that they were mainly composed of filamentous cyanobacteria probably belonging to the genus *Leptolynbya* (Figure 5). In area SPB-2, some coccal green algae were also observed. These cyanobacteria are often present in confined environments [26] where they thrive in extreme conditions at low light.

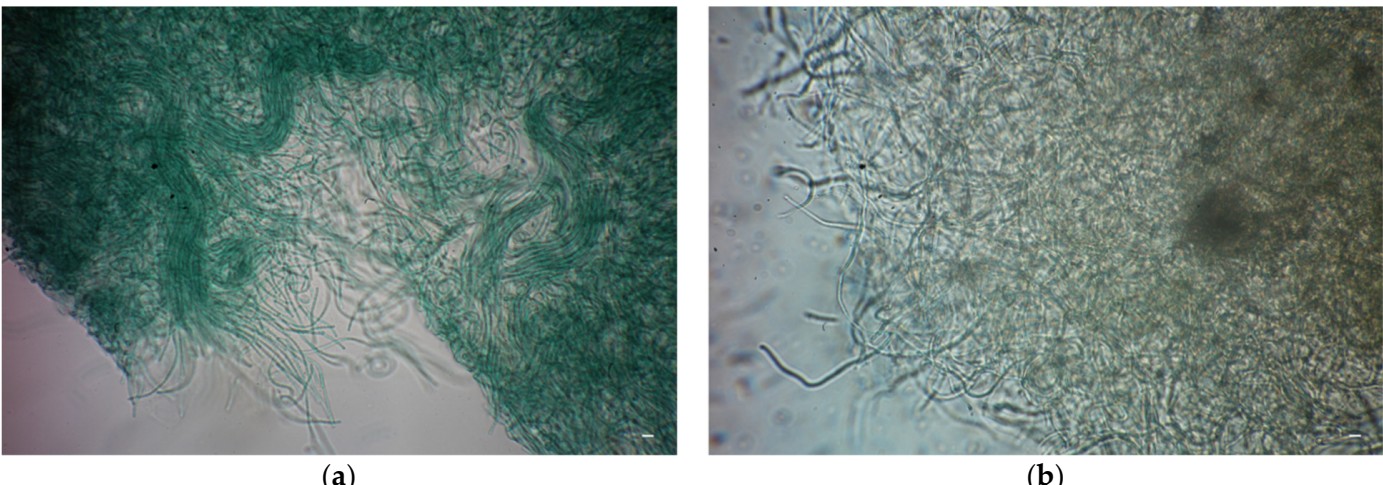

(**a**)                                                                                                    (**b**)

**Figure 5.** Optical images at 40× of SPB-1 (**a**) and SPB-2 (**b**) areas of the *Putridarium* in the church San Pietro Barisano, before the cleaning procedures. Bar = 5 μm.

Once the SPB-1 and SPB-2 areas were characterized, the BH_1 hydrogel was applied by brush with the cotton gauze on the respective surfaces. As just introduced, the cotton gauze allows uniform application and facilitates subsequent removal. As the environmental relative humidity of the hypogeum was very high, the hydrogel was not completely dried after 24 h; consequently, it was not possible to remove the cotton gauze unless using a thermo-convector.

Following the cleaning procedure for SPB-1 and SPB-2, the biofilms appeared effectively removed, as shown in Figure 4b,e,g. To evaluate the effectiveness of the cleaning, both areas were analyzed in situ with the digital microscope; the images confirm an efficient removal of microorganisms (see Figure 4e,j) and the absence of any residual cleaning components on the surfaces.

After cleaning, the SPB-1 and SPB-2 areas were ideally divided into three equal parts named NT, AC012 and P2. The NT subareas were left untreated, while on the AC012 and P2 surfaces, protective treatments with aqueous emulsions of Polyrest AC012 and Polyrest P2 were carried out, applied twice by brush until imbibition. After 24 h from the first application of Polyrest AC012, the stone substrates were still very brittle and tended to dust; on the contrary, the areas treated with Polyrest P2 were more compact just after the first application. With the second application, all the treated surfaces appeared well consolidated and much less friable of the corresponding NT areas, and any visible and unaesthetic bidimensional polymeric film was observed.

Finally, for testing the long-term effectiveness of the cleaning and protective procedures in SPB-1 and SPB-2 areas, new photos (Figure 4c,h) were acquired two years later. Furthermore, some samples were collected by applying the adhesive tape on the stone surface and allowed to grow in nutrient culture medium to ensure that there was no development of microorganisms not visible to the naked eye. As shown in Figure 6, for both areas, the observations of samples collected two years after the treatments (Figure 6a,b) confirmed the absence of biological colonization, and no growth was observed in culture. However, in the part of SPB-1 close to the limit of the treated area, which was affected by a green patina (Figure 6c), few hormogonia of the cyanobacterium *Leptolyngbya* were present (Figure 6d), indicating a beginning of colonization by the neighboring parts towards the treated area.

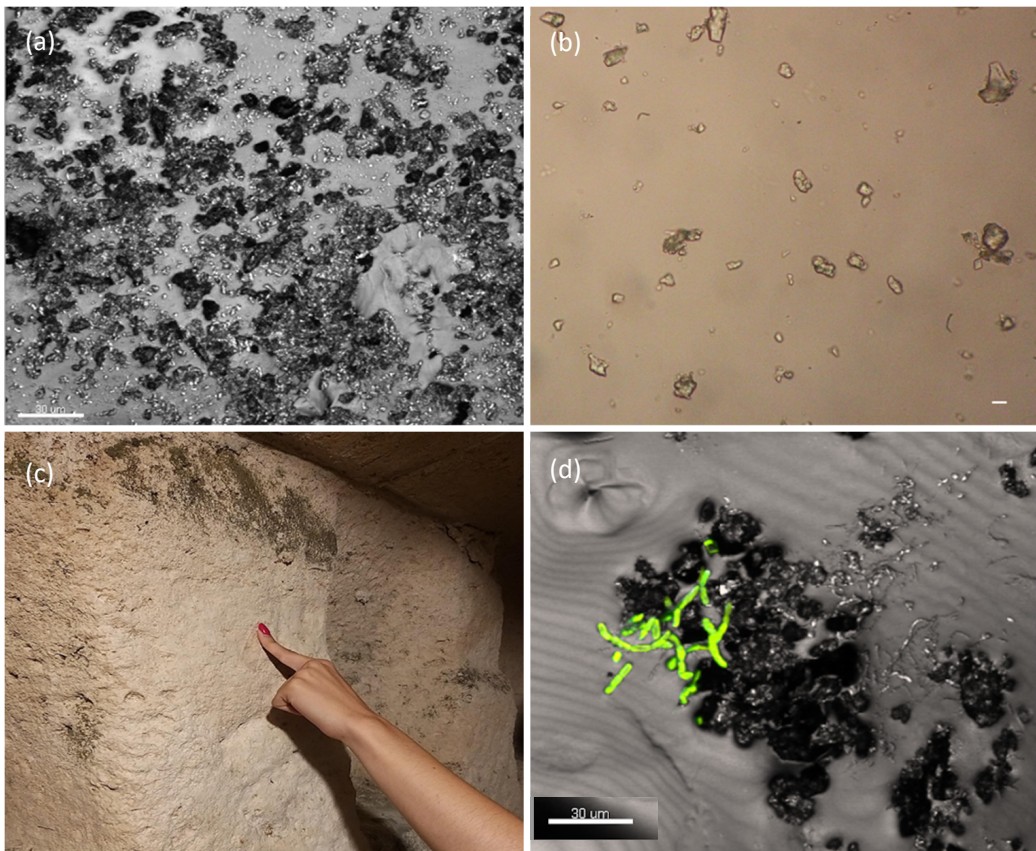

**Figure 6.** Images obtained with CLSM (**a**) and light microscope (**b**) after the cleaning procedure by observation of the adhesive tape strips of samples collected respectively at SPB-1 and SPB-2 areas (bar = 5 µm). (**c**) Image of sampling point in area SPB-1 with an evident difference between the colonized untreated upper area and the treated lower area; (**d**) CLSM image of the adhesive tape strips collected in the treated area at the point shown in (**c**).

As previously anticipated, the colorimetric analysis was carried out on both areas, before the cleaning procedure (COLONIZED), shortly after the cleaning procedure on the nontreated subareas (CLEANED NT) and shortly after the cleaning and protective procedures on the subareas treated with the coating products (CLEANED AC012 and CLEANED P2). Finally, colorimetric analysis was carried out on all the cleaned and eventually protected subareas, after two years (CLEANED 2y NT, CLEANED 2y AC012 and CLEANED 2y P2). All data, reported in Table 1, allow us to compare, in each phase of our restoration activity, the aesthetic properties of SPB-1 and SPB-2 with those pertaining to REF-SPB, the reference area previously determined.

From the analysis of the $\Delta E^*$ values, a large difference can be seen between the reference and colonized areas, while a minimal variation is detectable by comparing the reference with the cleaned areas. The color variation can be considered imperceptible to the naked eye when $\Delta E^* < 5$ [27]. As shown in Table 1, both protective coatings did not alter the chromatic aspect of the treated surfaces, and the long-term tests were fully satisfying.

**Table 1.** Mean chromatic coordinates of SPB-1 and SPB-2 areas, before, shortly after and 2 years after the cleaning procedure. For cleaned and two years later data, the values refer to subareas: nontreated (NT) and treated with protective coating Polyrest AC012 (AC012) or Polyrest P2 (P2). The corresponding mean color difference, $\Delta E^* = ((\Delta L^*)^2 + (\Delta a^*)^2 + (\Delta b^*)^2)^{1/2}$, references the chromatic coordinates of REF-SPB.

| **REF-SPB** | | **L\*** | **a\*** | | **b\*** | |
|---|---|---|---|---|---|---|
| | | 72 ± 5 | 4.8 ± 0.9 | | 13 ± 2 | |
| | | **ΔL\*** | **Δa\*** | **Δb\*** | **ΔE\*** | |
| | **COLONIZED** | - | −28.0 | −4.8 | 3.3 | 29 |
| | **CLEANED** | NT | −4.6 | 0.1 | 2.2 | 5 |
| | | AC012 | −1.9 | 1.9 | 2.4 | 4 |
| **SPB-1** | | P2 | −4.7 | −0.5 | 1.7 | 5 |
| | **CLEANED 2y** | NT | −4.6 | 1.7 | 2.0 | 5 |
| | | AC012 | −3.9 | 0.9 | 0.9 | 4 |
| | | P2 | −3.1 | 0.1 | 0.7 | 3 |
| | **COLONIZED** | - | −28.4 | −3.4 | 0.8 | 29 |
| | **CLEANED** | NT | 3.6 | 0.5 | 2.8 | 5 |
| | | AC012 | −3.9 | 0.4 | 1.5 | 4 |
| **SPB-2** | | P2 | 0.9 | 0.1 | 2.0 | 2 |
| | **CLEANED 2y** | NT | 0.9 | 0.3 | 0.3 | 1 |
| | | AC012 | 2.4 | 0.4 | 0.6 | 3 |
| | | P2 | 0.3 | 0.1 | 0.5 | 1 |

*3.2. Site 2: Madonna dei Derelitti Church*

The environmental parameters of the church Madonna dei Derelitti are very different from the previous site. The church is closed to tourists, there is no artificial light and the internal lighting and exchange with the exterior are guaranteed by the presence of a grid in the entrance portal. Furthermore, inside the small church, even if excavated in the limestone in its deepest part, temperature and humidity are very similar to the external ones. In particular, on the days in which the treatments were carried out, the daytime temperature varied between 15 and 20 °C with a humidity of around 70%–80% and the light irradiance inside the church ranged between 1 and 2 mol of photons $m^{-2} \cdot s^{-1}$.

The interior of the church presented various contaminations and states of decay; for this reason, different intervention areas were identified according to the type of degradation. The MD-1, MD-2 and MD-3 areas were representative of the main degrees of colonization observed within the church. In order to evaluate the effects of the treatment with alginate hydrogel supplemented with NaDCC (BH_2) and the protective coating products, Polyrest AC012 and Polyrest P2, photographic documentation and colorimetric and peeling tests were carried out.

Figure 7 shows the images of the three selected areas before, shortly after and two years after the cleaning treatment.

The MD-1 area (Figure 7a), located to the left of the altar, had a dark green biopatina that probably was in a fairly advanced state. Nevertheless, only one application of BH_2 was enough to obtain its effective removal, as displayed in Figure 7b.

The MD-2 area (see Figure 7d), located to the right of the altar, had a lighter green color than the previous one. The treatment with HB_2 was particularly effective on MD-2 after only one application, as shown in Figure 7e.

**MD-1**

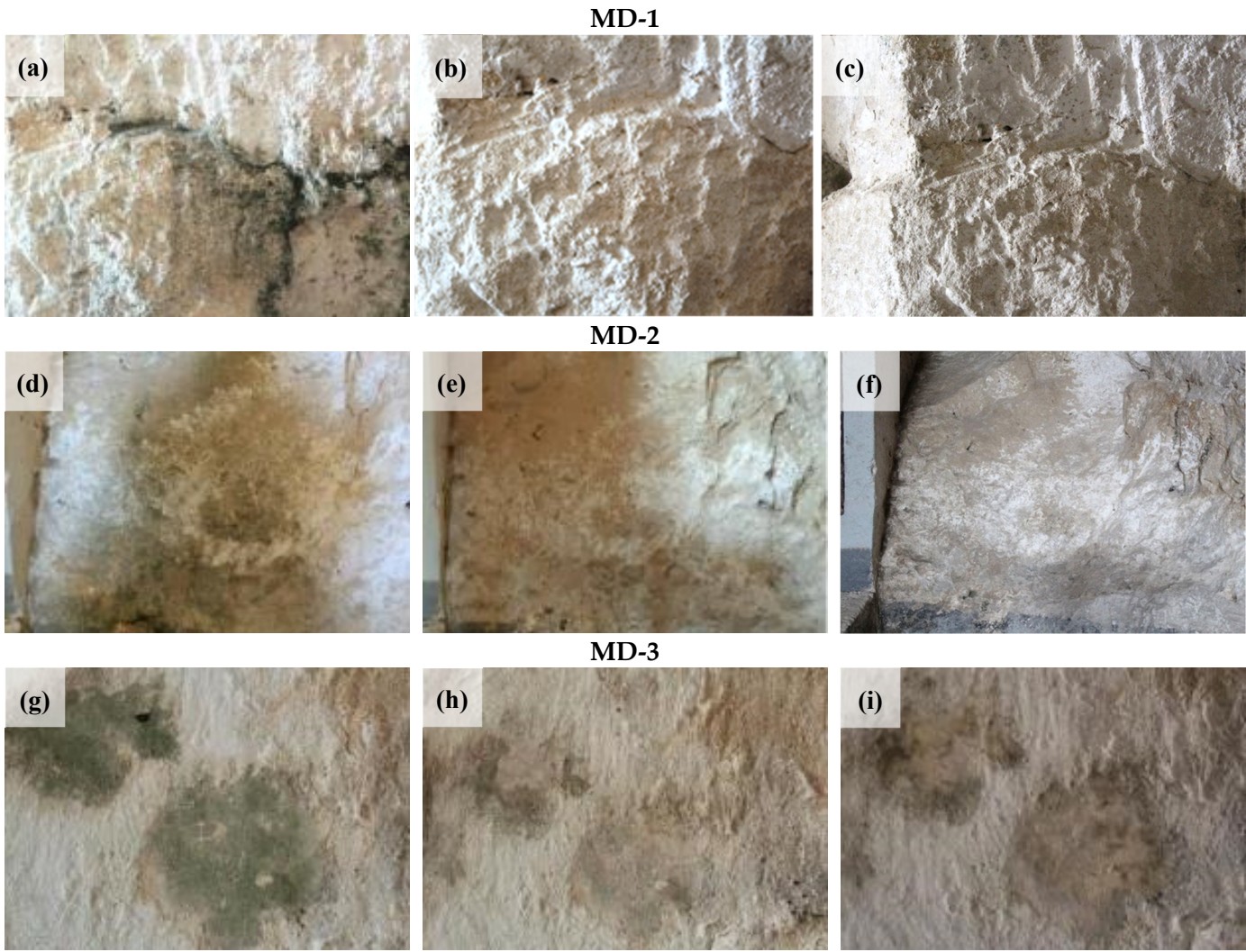

**Figure 7.** Photos of MD-1 (top row), MD-2 (middle row) and MD-3 (bottom row) inner areas of the church Madonna dei Derelitti, before (**a,d,g**), shortly after (**b,e,h**) and two years after (**c,f,i**) the cleaning procedure.

On the contrary, a single HB_2 treatment was not effective on the area called MD-3 (Figure 7g), and it was hypothesized that the microorganisms were incorporated into a carbonate matrix following the whitewashing. Consequently, before applying the biocide a second time, alginate/acetic acid gel (10 and 6% respectively) was applied to promote decarbonation in a weakly acid environment and, by means of the alginate chelating matrix, to sequester calcium ions. Following this pretreatment, BH_2 was applied again, and most of the biological patina was removed, as shown in Figure 7h.

As expected, the microorganisms responsible for the colonization of the church Madonna dei Derelitti were quite different from those found in the church San Pietro Barisano due to the completely different environment. As shown in Figure 8, the patinas were formed by cyanobacteria belonging to the genus *Chroococcus* or *Gloeocapsa* (Figure 8a); fungi (Figure 8b); green microalgae belonging to the genus *Muriella* (Figure 8c); and diatoms, probably a species of *Humidophila* sp. (Figure 8d). Figures S2–S4 in the Supplementary Materials show some selected SEM images of reference, colonized and cleaned fragments collected inside the church along with EDS spectra. Table 2 reports the weight percentages of the main elements present in several small samples taken from both sides of the altar, i.e., MD-1 and MD-2.

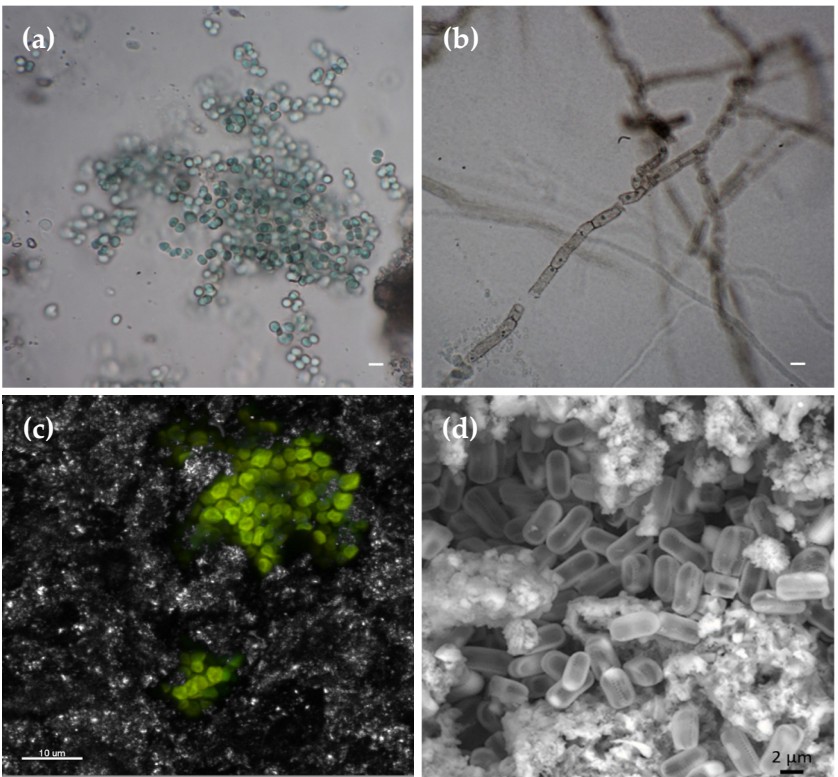

**Figure 8.** Images of samples collected at different areas of the church Madonna dei Derelitti, before the cleaning procedures obtained after observations by (**a**,**b**) light microscope (bar = 5 μm, 40×), (**c**) CLSM and (**d**) SEM.

**Table 2.** EDS elemental analysis of reference (REF), colonized and cleaned fragments collected at MD-1 and MD-2 areas.

| Element | REF | COLONIZED | CLEANED |
|---|---|---|---|
| O | 48 ± 3 | 35 ± 9 | 47 ± 2 |
| Ca | 35 ± 3 | 11 ± 3 | 37 ± 5 |
| C | 13 ± 3 | 33 ± 7 | 13 ± 3 |
| Cl | 1.0 ± 0.3 | 2 ± 1 | 0.8 ± 0.3 |
| S | 0.8 ± 0.2 | 0.5 ± 0.4 | 0.6 ± 0.2 |
| Si | 0.8 ± 0.3 | 16 ± 7 | 0.8 ± 0.5 |
| Na | 0.6 ± 0.3 | 0.3 ± 0.1 | 0.5 ± 0.1 |

As highlighted by the data in the table, the colonized areas showed a significant increase in both Si and C compared to the reference samples, both indicating microbial contamination and in particular the silica shell of diatoms. After the cleaning treatment, the percentages of the main elements were very similar to the reference ones.

All these microorganisms were described on rocky substrates in natural caves and sub-aerial habitats [28–30] and were already reported inside the "Crypt of the Original Sin", an exceptional frescoed rock oratory located 10 km south of Matera [31,32]. Moreover, a study on fungal contamination in the church Madonna dei Derelitti revealed the presence of several genera, such as *Parengyondontium*, *Alternaria*, *Cladosporium* and *Verticillium* [33].

After cleaning and the acquisition of colorimetric parameters, the selected areas were subjected to protective treatment; in particular, for the MD-1 area, the aqueous emulsion of Polyrest AC012 was used, while Polyrest P2 was applied on the MD-2 and MD-3 areas, always twice by brush until imbibition.

Two years later, all the surfaces subjected to both cleaning and protective treatments did not show any visual modification to suggest any kind of microbial colonization (Figure 7c,f,i).

In addition, in this case, several samples were collected by means of the adhesive tape strips on the three selected areas, and both the direct observation by light microscopy and CLSM and the growth in agarized culture medium proved the absence of biological colonization (Figure S5).

In order to evaluate any variations in the stone properties after the treatments with the biocidal hydrogel and the protective coatings, there was the need to identify uncolonized areas to be used as reference. The high color variability, visible with the naked eye and due to the alternation of live rock and lime due to whitewashing processes, made it difficult to choose a common reference area.

As highlighted by the colorimetric analysis in Table 3, REF-MD-1/2 was selected as a reference both for MD-1 and MD-2 areas, while REF-MD3 was chosen for MD-3.

**Table 3.** Mean chromatic coordinates of MD-1, MD-2 and MD-3 areas, before, shortly after and 2 years after (2y) the cleaning procedure. For cleaned and two years later data, the values refer to subareas: treated with protective coating Polyrest AC012 (AC012) or Polyrest P2 (P2). The corresponding mean color difference, $\Delta E^* = ((\Delta L^*)^2 + (\Delta a^*)^2 + (\Delta b^*)^2)^{1/2}$, references chromatic coordinates of REF-MD-1/2 for MD-1 and MD-2, while REF-MD-3 is the reference for MD-3.

| REF-MD-1/2 | L* | | a* | | b* | |
|---|---|---|---|---|---|---|
| | 67 ± 4 | | 5 ± 1 | | 12.0 ± 0.9 | |
| | | $\Delta L^*$ | $\Delta a^*$ | $\Delta b^*$ | $\Delta E^*$ | |
| MD-1 | COLONIZED | −36.1 | −3.8 | −6.4 | 37 | |
| | CLEANED | −0.6 | −0.5 | 2.2 | 2 | |
| | AC012 TREATED | 1.0 | −1.7 | 0.8 | 2 | |
| | AC012 TREATED 2y | 2.7 | −1.2 | 0.1 | 3 | |
| MD-2 | COLONIZED | −29.2 | −4.2 | −4.7 | 30 | |
| | CLEANED | 3.8 | −1.2 | −2.9 | 5 | |
| | P2 TREATED | 2.5 | −1.7 | −1.6 | 3 | |
| | P2 TREATED 2y | 2.4 | −1.2 | −1.3 | 3 | |
| REF-MD-3 | L* | | a* | | b* | |
| | 76 ± 2 | | 3.4 ± 0.61 | | 8.6 ± 0.9 | |
| | | $\Delta L^*$ | $\Delta a^*$ | $\Delta b^*$ | $\Delta E^*$ | |
| MD-3 | COLONIZED | −13.7 | −6.5 | 0.8 | 15 | |
| | CLEANED | −2.7 | −2.3 | −1.0 | 4 | |
| | P2 TREATED | 0.0 | −1.2 | −1.1 | 1 | |
| | P2 TREATED 2y | −3.4 | 0.1 | 2.1 | 4 | |

Colorimetric parameters acquired on both MD-1 and MD-2 areas before the cleaning procedure showed an abrupt decrease in brightness and a significant variation in the colorimetric parameters a* and b* compared to their reference. Following the cleaning with HB_2, the MD-1 area presented a yellowing, as indicated by the variation of b* parameter, probably induced by the mechanical removal of part of the more damaged and less adherent lime present on the stone support.

Instead, no chromatic alterations related to treatment with the hydrogel are noted in the MD-2 area; the differences in brightness and in color were negligible if we consider the chromatic variety that characterizes the building material.

As regards the MD-3 colonized area, data in Table 3 indicated a decrease in L* and a* parameters with respect to the reference area, but an almost equal b* value. As in the previous cases, the cleaning procedure restored the chromaticity of the reference surface. Moreover, the colorimetric data relating to the areas treated with the protective products,

acquired shortly after the treatment and two years later, showed excellent agreement with their respective references.

In summary, the color difference $\Delta E^*$, very evident in colonized areas, was drastically reduced after the removal of the biopatina by the hydrogel and remained below the threshold value after the coating application. Moreover, treatments proved to be stable over time compared to the variability of the chromatism that characterizes the interior of the church.

Finally, peeling tests were performed on at least five different zones of MD-1 and MD-2 areas, before and after the treatment with the protective products, to evaluate the reduction in chalking thanks to the application of the coatings (Table 4).

**Table 4.** Peeling test for MD-1 and MD-2 cleaned areas, before (CLEANED) and after the application of Polyrest AC012 or Polyrest P2 protective product.

|  |  | Weight of Detached Material, mg/cm$^2$ |
| --- | --- | --- |
| **MD-1** | CLEANED | $2.5 \pm 0.9$ |
|  | AC012 TREATED | $0.4 \pm 0.1$ |
| **MD-2** | CLEANED | $1.6 \pm 0.9$ |
|  | P2 TREATED | $0.2 \pm 0.1$ |

As can be seen from the data, a single protective treatment considerably decreased the chalking of the stone by about 83% for the subarea MD-1 AC012 TREATED with respect to MD-1 CLEANED and by about 91% for MD-2 P2 TREATED with respect to MD-2 CLEANED.

Several examples of in situ application of commercial products are reported in the literature, and many of them involve the use of quaternary ammonium salts, such as benzalkonium chloride [34–36] or isothiazoline-based biocides [34,35,37–39]. Despite their effectiveness in removing biological patinas, they must be used in high concentrations, and multiple cleaning treatments must often be performed; moreover, some bacterial strains can use quaternary ammonium salts as a source of carbon and nitrogen, inducing the formation of new biofilms [37]. To overcome these drawbacks, environmentally friendly cleaning methods are being developed, such as microwave heating systems [36,39], UV-C treatments [38] and the use of biocides of plant origin (alcoholic extracts or essential oils) [22].

The cleaning procedure developed in this study is based on the use of two inexpensive oxidizing biocides no longer used in the field of cultural heritage due to the side effects they cause when used in solution. In fact, to obtain the removal of the biopatinas, they must be used at high concentrations with a consequent degradative effect on the artifact surface, such as salt efflorescence or corrosion. Their encapsulation on a gel matrix has allowed us to reduce the amount of oxidizing agents, preventing the penetration of hygroscopic salts inside the stone materials while not modifying the chromaticity of the treated supports, and minimize their impact on the environment and on the operators. Furthermore, the application of acrylic protective coatings after the biocide treatment showed the absence of recolonization two years after the restoration.

## 4. Conclusions

Biodeterioration due to the development of microorganisms on stone monuments is a widespread problem that endangers most cultural heritage sites, and great efforts are needed to find sustainable, safe and effective solutions. Previous laboratory experience on new effective biocidal hydrogels allowed us to use a reduced amount of biocide in a treatment process with easy application and removal for the full restoration of the hygroscopic and colorimetric properties of biodeteriorated substrates. These results enabled the cleaning procedure to be transferred to in situ application on some internal areas of two rupestrian churches belonging to "The Sassi and the Park of the Rupestrian Churches of Matera". Despite the differences both in environmental conditions and in the microorgan-

isms responsible for biodeterioration, the application of these biocidal hydrogels resulted in the complete removal of the biopatinas and the restoration of the original chromaticity, as shown by colorimetric measurements and microscopic investigations. Moreover, after two years, the action of cleaning and protective treatments was successfully confirmed, indicating the long-term efficacy of the proposed biocleaning procedure.

**Supplementary Materials:** The following supporting information can be downloaded at: https://www.mdpi.com/article/10.3390/coatings12040462/s1, Figure S1: (A) FTIR-ATR spectra in the region comprise between 1500 and 900 cm$^{-1}$ and (B) DSC thermograms of 5 wt.% alginate hydrogel (——), BH_1 (——) and BH_2 (——); Figure S2: SEM images of reference stone materials fragments collected inside the Church of the "Madonna dei Derelitti" and EDS analysis of two selected areas; Figure S3: SEM images of colonized stone materials fragments collected inside the Church of the "Madonna dei Derelitti" and EDS analysis of two selected areas; Figure S4: SEM images of stone materials fragments collected after cleaning treatment inside the Church of the "Madonna dei Derelitti" and EDS analysis of two selected areas; Figure S5: Images of samples collected at different areas of the Church of "Madonna dei Derelitti", after the cleaning procedures obtained after observations at (a) the light microscope (bar = 5 μm, 40×) and (b) CLSM. It is evident the absence of living microorganisms; mineral material and cellular debris are visible in (a) among which empty frustules of diatoms (arrow); Table S1: Summary of the investigated properties of pure alginate hydrogel (HY) and biocidal gels containing calcium hypochlorite (BH_1) and sodium dichloroisocyanurate (BH_2). References [40–43] are cited in the supplementary materials.

**Author Contributions:** Conceptualization, L.B., C.C. and N.S.; investigation, F.G., R.R. and L.R.; visualization, F.G.; writing—original draft, F.G., C.C. and N.S.; writing—review and editing, L.B., C.C. and N.S.; funding acquisition, C.C. and N.S.; supervision, N.S. All authors have read and agreed to the published version of the manuscript.

**Funding:** This work was supported by the Ministry of Education, Universities and Research (MIUR): project Smart Cities and Communities and Social Innovation on Cultural Heritage (SCN_00520).

**Institutional Review Board Statement:** Not applicable.

**Informed Consent Statement:** Not applicable.

**Data Availability Statement:** Not applicable.

**Acknowledgments:** We are also grateful to Antonella Guida and Antonello Pagliuca (University of Basilicata) for their support in the in situ trials. The authors are indebted to Chiara Gandolfi (Icap Leather Chem SpA, Milan, Italy) for supplying the experimental products. The authors thank the restaurateur Luca Pantone of Pantone Restauri Srl (Rome, Italy) for precious assistance during the in situ activities and Maria Giammatteo and Lorenzo Arrizza (Microscopy Centre, University of L'Aquila) for their kind assistance in SEM/EDS analysis.

**Conflicts of Interest:** The authors declare no conflict of interest.

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
