# Peer review of "Application and Monitoring of Oxidative Alginate–Biocide Hydrogels for Two Case Studies in “The Sassi and the Park of the Rupestrian Churches of Matera”"

_coatings, doi:10.3390/coatings12040462_

Round 1
Reviewer 1 Report
The submitted article for review corresponds to the subject of the authoritative scientific journal Coatings (ISSN 2079-6412).
Application and monitoring of oxidative alginate-biocide hydrogels for two case studies in “The Sassi and the Park of the Rupestrian Churches of Matera”
The title is good and quite specific and attracts the reader's interest so that it is easy to understand.
The topic of scientific research is quite interesting and relevant.
The scientific article is logically built, corresponds to the principles of presenting scientific information and research.
I think that the article used enough tables, graphics and illustrations.
I note that the authors of the article treated the analysis of the problem quite thoroughly and used the necessary number of sources of information to prepare the submitted manuscript.
Author's notes
Abstract
Abstract looks very general. Authors should mention the importance of research work briefly. Add more discussion related to the research, values ​​of research output and future research directions as well.
Add a few more keywords.
- Materials and Methods
Please clarify - on the basis of what document were scientific research allowed at cultural heritage sites?
Lines 119-121, 124-126.
Please add - indicate the date (month and year) of the beginning of the research and the date (month, year) of the end of this study.
Lines 178-183
I consider it unacceptable to evaluate commercial materials in comparison with analogues in this section of a scientific article.
- Results and Discussion
Line 223
Figure 3. b) digital microscopy – is it possible for the authors to present a clearer image?
References
I was unable to validate references #27 (line 489).
I recommend for publication the scientific article "Risk assessment of heavy metals contamination in pork" - FOODCONT-D-21-01129. Application and monitoring of oxidative alginate-biocide hydrogels for two case studies in “The Sassi and the Park of the Rupestrian Churches of Matera”(coatings-1629197) after the comments have been corrected.
Prof. Dr. Maksim Rebezov
V.M. Gorbatov Federal Research Center for Food Systems of Russian Academy of Sciences, Moscow, Russian Federation
https://elibrary.ru/author_profile.asp?id=419764
http://orcid.org/0000-0003-0857-5143
https://www.scopus.com/authid/detail.uri?authorId=54974244300
https://publons.com/researcher/2287875/maksim-rebezov/

Author Response
- Abstract looks very general. Authors should mention the importance of research work briefly. Add more discussion related to the research, values ​​of research output and future research directions as well
Abstract has been deeply modified.
- Add a few more keywords
Some other keywords have been added.
- Please clarify - on the basis of what document were scientific research allowed at cultural heritage sites?
A wide spread of multidisciplinary results was obtained for the Smart Cities and Communities and Social Innovation project (SCN_00520) supported by MIUR and entitled “Product and process innovation for the maintenance, conservation and sustainable restoration of cultural heritage”. The project basically suggested the opportunity to avoid emergency closures of sites of high historical, architectural and cultural value, to proceed with planned interventions that regularly guarantee the well-being of the site and its conservation. A new organization of the protection, conservation and maintenance system was thus proposed. Monumental heritage of Southern Italy and in particular in ‘The Sassi and the Park of the Rupestrian Churches of Matera – UNESCO World Heritage Site” was the site where part of the lab activity was transferred “in situ”. In particular, the reported activity was aimed at ensuring a sustainable restoration based on safe cleaning methods on calcareous stones, able to reduce professional diseases and environmental pollution. Thanks to expression of interest of the local authorities, responsible for the protection of the archaeological park, we obtained the necessary permits for the activity here reported. To clarify this concept a short sentence at begin of 2.1 paragraph has been added.
- Lines 119-121, 124-126: Please add - indicate the date (month and year) of the beginning of the research and the date (month, year) of the end of this study
The dates of the beginning and of the end of this study have been added.
- Lines 178-183: I consider it unacceptable to evaluate commercial materials in comparison with analogues in this section of a scientific article
The authors are sorry for this misunderstanding, but the comparison between the new coatings and others, just in commerce, is not argument of this paper. This work was fully reported in Ref. 21. The paragraph 2.7 has been modified.
- Line 223 Figure 3. b) digital microscopy – is it possible for the authors to present a clearer image?
A clearer image of Figure 3 b) has been inserted.
- I was unable to validate references #27 (line 489)
Reference #27 has been modified.
Reviewer 2 Report
This paper on the protection of cultural assets contains significant findings.
The condition of the treated area after two years was also reported, indicating from the paper that the treatment was well applied. It was decided that this paper could be accepted with only a brief additional explanation.
・What kind of polymeric film is a "polymeric film"?
Author Response
- What kind of polymeric film is a "polymeric film"?
For what concerns coatings for support materials pertaining cultural heritage, it is apparent that esthetics are of great importance, and consequently influence selection of the raw materials to be used in the coating formulation. With the generic term 'polymeric film' we mean the undesirable formation of a clearly visible surface layer of the protective product that distorts the aesthetic properties of the substrate.
Reviewer 3 Report
Dear authors,
This paper deals with the application and monitoring of oxidative alginate-biocide hydrogels for two case studies.
This is an interesting paper devoted to applying a biocide hydrogel in real life restoring sites. However, several issues must be overcome before being considered for publication.
The abstract must include quantitative information and must be improved giving information of coating used
Methodology. It is suggested to change order of presentation of information...e.g. 2.1, 2.2, 2.5, 2.3.
Results section. The data about the characterization of biocidal hydrogel must be provided and included in the manuscript... or in the supplementary information. (e.g. FTIR, TGA, elemental analysis, viscosity).
More quantitative data is necessary to be provided. Does the sampling of sites were made only at year 2. If previous sampling was performed, please provide data to understand the evolution of the site composition along time.
I suggest the SEM-EDX analysis would be included and discussed inside the manuscript and samples composition would be presented in a Table.
Peeling test statistical analysis must be performed and discussed.
A better result discussion must be prided including comparison with other authors work.
Conclusions must be improved.
Author Response
- The abstract must include quantitative information and must be improved giving information of coating used
Abstract has been deeply modified.
- It is suggested to change order of presentation of information...e.g. 2.1, 2.2, 2.5, 2.3
The order of the paragraph in Material and Methods has been changed.
- The data about the characterization of biocidal hydrogel must be provided and included in the manuscript... or in the supplementary information. (e.g. FTIR, TGA, elemental analysis, viscosity)
The characterization of biocidal hydrogel has been reported in Supplementary Materials.
- More quantitative data is necessary to be provided. Does the sampling of sites were made only at year 2. If previous sampling was performed, please provide data to understand the evolution of the site composition along time
Effectively, it was our intention to monitor the treatments several times over two years, but the Covid19 emergency prevented us. Nevertheless, the longtime good results allow us to suppose a positive behavior along the two years.
- I suggest the SEM-EDX analysis would be included and discussed inside the manuscript and samples composition would be presented in a Table
As suggested by the Reviewer, the samples composition by SEM-EDS analysis has been included in Results and Discussion on the basis of a new Table.
- Peeling test statistical analysis must be performed and discussed
As specified at old line 399, we performed, for each of the four areas, five sampling in different zones. Given the low number of samples, a deeper statistical analysis would not have been appropriated, so that, the errors were simply calculated as maximum semi-dispersion.
- A better result discussion must be provided including comparison with other authors’ work
Discussion has been integrated with a comparison with other recent References about removal of biopatina.
- Conclusions must be improved
Conclusions have been modified.
Round 2
Reviewer 3 Report
No coments